# An Institutional Analysis of Local Lifelong Learning Approaches to Early School Leaving in Italy and Spain

**Xavier Rambla** [1,*] and **Maddalena Bartolini** [2]

1    Department of Sociology, Universitat Autònoma de Barcelona, 08193 Cerdanyola del Vallès, Spain
2    IRCrES—Research Institute on Sustainable Economic Growth of the National Research Council of Italy, 10135 Torino, Italy; maddalena.bartolini@icres.cnr.it
*    Correspondence: xavier.rambla@uab.cat

**Abstract:** In this paper, we will analyze how a few initial vocational education and training (VET) schools have elaborated wide-ranging responses to early leaving from education and training in Liguria (Italy) and Catalonia (Spain). In contrast with many members of the EU and the OECD, the prevailing institutional arrangements in these two countries hardly support disadvantaged youths to catch up with basic academic performance and find an appealing pathway between compulsory and post-compulsory education. Despite this bias of official policies, we argue that some initial VET schools manage to navigate the interface between social structure and agency and accommodate the aspirations of disadvantaged students by involving local stakeholders and shaping school time organization in a particular way.

**Keywords:** VET schools; public policy analyses; early school leaving from education and training





## 1. Introduction

Our analysis draws on previous research about secondary education students' trajectories, VET policies and the analysis of public policies through the lens of historical institutionalism and coalition formation. We inquire why a few initial VET schools have developed wide-ranging responses to early leaving from education and training that aim at protecting young people from the perverse consequences of dropping out without an academic credential. This research question is very significant in countries where, traditionally, official policies have prioritized academic selection and disregarded support to the most disadvantaged, such as Italy and Spain.

Across the world, the relevance of early school leaving has changed as the education policy agenda has increasingly taken lifelong learning into account. Thus, the Sustainable Development Goals require that education continues after the end of compulsory schooling to "ensure inclusive and equitable quality education and promote lifelong learning opportunities for all" [1]. In a very similar vein, the European Pillar of Social Rights establishes that "everyone has the right to quality and inclusive education, training and life-long learning in order to maintain and acquire skills that enable them to participate fully in society and manage successfully transitions in the labor market" [2]. In the USA, the Every Student Succeeds Act (ESSA) also takes into consideration students' college prospects and further careers [3].

Thus, not only is early school leaving considered a painful consequence of academic underperformance but also a wicked problem in the long run. Of the considerable perverse effects, rates of striking are high in the Southwestern regions of the European Union, not least because official policy frameworks have traditionally overlooked the connections between compulsory education and lifelong learning. Nonetheless, in Liguria (Italy) and Catalonia (Spain), a few initial VET schools have become able to cater to the young people who cannot accredit the basic learning outcomes. The findings of previous research have triggered our curiosity on how the teachers and the students at these schools have

abandoned common practices in the country at the same time as having borrowed the international message about lifelong learning.

The two theoretical sections of the article associate a few spurs of change with the making of small, weak but relatively effective local coalitions around VET schools. A methodological section explains on which grounds the re-analysis of pieces of previous research sheds new light on the research questions and may hopefully indicate frontiers of our current sociological knowledge. Two further sections present the evidence collected and analyzed in Liguria (Italy) and Catalonia (Spain). A discussion section retrieves the theoretical argument to spell out the clues of inertia and transformation in these regions.

## 2. Institutional Modes of Weak Support across Life Course Regimes

Our research question elaborates on the literature about youth transitions, early school leaving and political analysis. We inquire about how some initial VET schools tackle early school leaving by encouraging former leavers to become protagonists of their education amid path-dependent institutional contexts that stymie the agency of young people. Our point of view is that this question contributes to the literature on national regimes of youth transition by inviting researchers to explore social interactions between supra-national, national and local policy actors.

Several researchers have convincingly argued that modern societies construe youth as a transition along institutional itineraries that involve 16- to 30-year-olds in various forms of education, training, employment and housing that correspondingly allow them to carry out their life plans differently. Everywhere, the outcome of older arrangements and ulterior policy changes configures structures of opportunities that make unequal life chances acceptable for young people [4]. Sociological discussions of agency and structure clearly resonate with this widely shared understanding of youth studies.

When the European Union advised the member states to prevent early school leaving and protect dropouts from the resulting damage in 2011, some observers noticed that many governments reacted reluctantly [5]. In fact, the record of this reaction confirms the finding that "support across life course regimes" is disparate across the European Union because the member states rely on different notions of disadvantage. While Scandinavian countries have established universal protection against the disadvantages of young people, Anglo-Saxon countries normally push them to enter the labor market as soon as possible, and Central European countries have developed a system of hierarchical protection that channels them toward uneven opportunities of employment [6].

Universalistic regimes provide support to disadvantaged youth within regular school or training. Anglo-Saxon regimes aim at instilling attitudes of work readiness in them. In Central Europe, support is centered on employment insofar as educational authorities, employers, unions and public services endeavor to enroll all young people into large apprenticeship schemes. In contrast, a proper institutional scheme has not been designed in Mediterranean countries [6]. In Belgium, France, Greece, Italy and Spain, mainstream policies restrict economic support to the most disadvantaged and consider that the youth are still depending on their families [7].

Certainly, neither in Italy nor in Spain have educational authorities been diligent in tackling the drivers and alleviating the consequences of early leaving [8,9]. However, the observation of proactive vocational schools recommends caution against too quick generalizations.

The local actors of educational policymaking are far from passive recipients of national and supra-national scripts. On some occasions, local networks of educational institutions draw on student-centered pedagogies to bring early leavers back to education and training. Schools normally collaborate with the local civil society to build these networks [10].

Networks of high schools, vocational schools and other VET providers succeed in governing students' trajectories in the local context. Comparative research has identified a type of school in which teachers engage in formal and informal partnerships with parents' associations and other organizations of the local civil society [11]. VET providers can also

fashion the local landscape of education and training. Thus, an array of local programs has aligned municipal services with the work of vocational schools and the activity of the public employment service in the Nordic countries [12]. In Austria and Germany, regional and local governments regularly design and implement specific programs to guarantee that the youth can participate in wide-ranging apprenticeship schemes even though they live in fragile conditions of social vulnerability [13,14].

Another strand of the literature has added that collective agency contributes to shape youth transitions [15–17]. Some studies have critically scrutinized local partnerships of VET providers [18] in diverse countries as Spain [19] and Sweden [20]. The point is that varied patterns of cooperation and competition between local providers [21] as well as of initiatives to tackle local uncertainties [22] eventually mold local opportunity structures.

### 3. Spurs of Local Change in Italy and Spain

We explore path dependency and spurs of local change that broaden opportunities amidst institutional arrangements that reduce young people to the role of dependent receivers of parental support. This focus bridges the two previous strands of the literature. Although national regimes are strongly influential, it is noticeable that this strength ultimately relies on the alignment between national and local policy orientations. While the bulk of the research has documented coordinated action between these two geographical levels in the Scandinavian enabling mode and the Central European employment-centered mode of youth citizenship, some previous findings already report exceptions in Anglo-Saxon regimes, where local actors sometimes undertake actions that build more comprehensive (albeit weak and small) arrangements at the local level [16,23]. We wonder to what extent these exceptions also take place in countries in which, after school leaving age, students are tracked to different programs and certain VET programs are tailored to the alleged needs of the most disadvantaged. France illustrates this type of official approach [7] at the national level, but some regions have adopted institutional trajectories that significantly expand the range of support [24]. Some qualitative studies have also found examples of resilience at the local level in Mediterranean countries [25].

We split this general interest into two more precise research questions. First, we ask to what extent schools can build policy coalitions at the local level. Institutional analyses consider that some actors build coalitions despite their weakness when their routines are shaken by the requirements of different structural domains, such as the challenges education, employment and social welfare posit to disadvantaged young people [26]. Sociological theory has pointed out that agents equip themselves with a wider capacity to establish social relations and develop new ideas when they engage in collective action [27]. Policy studies have also noticed that negotiation and learning are conducive to changes [28]. Therefore, we investigate whether some VET schools and VET programs sometimes become local policy actors which are capable of widening the opportunities of the youth by redefining local spaces, although their influence may be weak and unstable. Remarkably, previous research has found different articulations of national regimes with local institutional arrangements that underpin the enabling [12] and the employment-centered [13] modes of youth citizenship as well as facilitate the agency of the youth despite the constraints posited by Anglo-Saxon [23] and Mediterranean regimes [24]. We will attempt to spell out the clues of some analogous processes in Italy and Spain, although the national approaches are extremely narrow.

Second, we look at VET schools and programs that cater for at-risk students in Liguria and Catalonia and seem to have shaped a meaningful mode of temporality for their students. The notions of transition, early leaving, trajectory and age group remind us of the intermingling scopes of youth studies and the sociology of time. Therefore, we also investigate whether some VET schools and programs are capable of molding the time of education and training in such ways that allow for young people to become more reflective, learn about the opportunities ahead and become protagonists of their choices. Crucial to

our argument is that besides configuring a networked local space, these schools also learn to experiment with multidimensional temporality.

Time studies normally distinguish "time" and "temporality". The measurement of minutes and hours instantiates a standard concept of time that is nonetheless a variety among others. The social construction of time (or temporality) refers to the wide array of understandings. A stylized but telling illustration has repeatedly contrasted the traditional temporality of life, harvest and ritual cycles with the modern time of clocks and timetables.

Schooling has typically relied on a standard concept of homogeneous school temporality that reproduces the movements of a clock [29]. Accordingly, teachers mostly read the results of evaluations in the short term and figure out new methods that improve academic performance in the near future [30]. Such linear temporality establishes a very influential social norm according to which teenagers grow up as they pass through academic years and cross through legal thresholds such as school leaving age [31]. Clearly, early school leavers are construed as a social problem insofar as their feelings of school disengagement often disrupt such a homogeneous school temporality, and their very existence challenges the assumption that compulsory education finishes at a certain age.

To the extent that they share a construal of linear time, many educators systematically overlook the increasingly diverse processes of the life course and the extension of young people's deliberation about future projects over longer periods than the last months of compulsory education [32]. The conception of early school leaving as a statistical stock of 18- to 24-year-olds produces a similar institutional bias insofar as this datum overlooks the intricacies of individual life courses. In contrast, several qualitative studies have reported that many youngsters leave school early when their prior school experiences intermingle with recurrent contradictions between family and school life, not least if they discover real opportunities to experiment with work that schools do not recognize [32,33]. Education, training and employment policies also shape particular images of school and educational times, which complicate the circumstances and the consequences of students' decisions at the threshold of the school leaving age [34,35]. Nevertheless, although the conception of homogeneous time normally prevails in most institutions, several schools, teachers and educators have noticed that education is an open-ended space of lifelong learning, where everybody may participate in education regardless of age [35].

In conclusion, lifelong learning approaches to early school leaving have become a significant object of sociological research. These approaches challenge the social norm that education is a right and an obligation of children and teenagers up to a certain age. In the European Union, the adoption of a right to lifelong learning intermingles with inherited understandings of the disadvantages that constrain young people. While these ideas endorse the agency of the youth in the north and the center of the continent, the alignment of structures and opportunities is more complicated southward. We inquire to what extent national narrow understandings of youths' social rights completely determine the action of VET schools in Italy and Spain. The literature suggests that these educational institutions may become policy actors insofar as they engage in coalitions with local authorities and non-profits that stand for more ambitious visions of lifelong learning and the corresponding actions against early leaving. Apparently, the members of such coalitions often deploy more suitable strategies at the same time as they build their alliances. Similarly, the literature suggests that these initial VET schools set multidimensional patterns of school temporality as they engage with the local reality. This point leads us to hypothesize as to what extent these patterns are meaningful for young early school leavers who look for new opportunities. In the following sections, we report how we can investigate these research questions.

## 4. Taking Geographical Levels into Account

Our analysis must inevitably look at national and local spaces if we are to spell out the contradictions of policies against early school leaving when official discourses assume that young people need parental rather than public support. We draw on the evidence

that previous studies collected in Italy and Spain between 2015 and 2018. Although the big picture is more complex, because Europe, member states, regional governments and municipalities have a say on these policies, for the sake of simplicity, our analysis will only take national and local scales into consideration.

We follow four steps to conduct this analysis. We start with general trends and then define what we understand by "case". Afterwards, we discuss how we selected the cases. Finally, we review the evidence that those previous studies provided for this new analysis.

First, the guidelines of the comparative method advise researchers to take general trends into consideration and compare cases in particular ways that are relevant for theory building [36]. In the introduction, we noticed a very large trend such as the growing importance of lifelong learning and the interest of key international organizations and governments to link lifelong learning with local and regional governance. Inasmuch as lifelong learning encompasses all types of education during the life of people, dealing with early school leaving lies at the heart of this concept as well as at the crossroads of the biographies of many young people.

Borrowing from the methodology of the social sciences, it is plausible to argue that our approach is abductive. Abduction consists of figuring out explanations of phenomena beyond the strict procedures of either induction or deduction [37]. Our investigation inquires whether local policy arrangements simply replicate national scripts in Italy and Spain. Even though the prevalence of extremely narrow definitions of the right to education does not invite us to expect exceptions in which schools underpin the agency of the youth, we contend that these exceptions exist and provide substantial food for thought.

Second, the guidelines of the comparative method problematize the concept of case [38]. In general, the cases of social and educational research comprise sets of actors that establish variable relationships between them. In this sense, we locate certain vocational schools and programs within local and regional contexts in which some policies are implemented. We are not describing these contexts in depth but taking them as a reference. This reference is important because the analysis particularly focuses on how teachers and students navigate an intricate landscape of policies and socio-economic opportunities in which students are construing their educational aspirations and eventually their life plans. Because the policies are not only the outcome of official decisions but also the medium of emerging networks of stakeholders, the cases are assemblages of heterogeneous pieces. If we look at them either from the top downwards or from the bottom upwards, in these landscapes, we identify many actors who are endowed with varied capacities.

A tentative list of actors, thus, should include teachers, students, policymakers, employers, non-profits and community associations. Our analysis focuses on the teachers and the students at some schools, the external actors with whom they interact and their capacities to shape space and time. Thus, the schools we observe in Liguria (Italy) and Catalonia (Spain) are complex cases in that heterogeneous social agents engage in interaction in these educational spaces. Our research considers several dimensions of policymaking (i.e., the European agenda, national implementation, local coordination) and the experience of students (i.e., their views on teachers, their expectations). The research also considers a range of geographical scales. This is a multidimensional and multilevel design [39].

Third, such multilevel design requires a two-pronged sequence of case selection that articulates national extreme cases with regional typical cases [40]. Italy and Spain are extreme cases in Walther's (2017) and Chevalier's (2016) typologies insofar as they have not articulated modes of economic and social support that aim at fostering the agency of the youth. However, Liguria and Catalonia are typical regions within these two states that have devolved some competencies of educational policy to regions.

On the one hand, in 2018, larger proportions than the EU 10% average had left school too early in Italy, Liguria, Spain and Catalonia [41]. Within the countries, both regions scored average rates, while the southern regions normally scored the highest rates.

On the other hand, in Italy and Spain, regional governments run parts of the educational system in a very similar way across the country. Actually, the main exceptions

are South Tyrol and Trentino-Alto Adige in Italy and the Basque Country in Spain, which are not in our sample. In Spain, all secondary schools are comprehensive, but only some of them deliver VET programs for students above 16 years old. In Italy, some secondary schools are specialized in academic programs, while the intake of other ones is mostly formed by disadvantaged students above 15 years old.

Finally, this analysis draws on evidence collected through corpuses of official documents as well as samples of interviews with policymakers, teachers and students involved in educational programs targeted to disadvantaged students in Liguria and Catalonia. We also use the transcripts of focus groups with students in both regions. The authors have been involved in research projects on this theme in each one. They designed the interviews and focus groups carried out in Liguria so that these social research techniques explored analogous research questions to the leading objectives of the projects carried out in Catalonia.

The evidence from Liguria comes from official documents, interviews and focus groups that were conducted in two vocational high schools that were experimenting with innovative methods to cater to the needs of students at risk of early leaving. Regional and local policymakers were included in the sample of interviews. Besides teachers, a few local educators were also interviewed outside the school. Students discussed their appraisal of the school in focus groups and were allowed to propose their own solution to the problems they noticed [42].

The evidence from Catalonia comes from research on early school leaving in high schools in the city of Barcelona, a research project on "second chance schools" in the metropolitan area of Barcelona, as well as a study on lifelong learning programs addressed to young adults in Girona. Many of the young interviewees were exposed to a severe risk of early school leaving [43–45]. Regional and local policymakers as well as teenagers and young adults were interviewed in the first and the third projects. The second project conducted focus groups with students who had enrolled in a second-chance school after leaving high school without any academic credential.

## 5. Governing Educational Pathways in Liguria (Italy)

In this section, we will analyze the policies that the government of Italy and the region of Liguria have implemented in order to improve the outcomes of initial VET since 2007. The starting point refers to an official plan. Because we draw on fieldwork carried out in 2017, the analysis looks at the pedagogies that schools were implementing at that moment.

During this period, the Italian central and regional authorities established new patterns of multilevel governance that address early school leaving by means of the European Social Fund (ESF) National Operational Plans (NOPs). These plans have compensated for the ravages of the austerity policies that responded to the European sovereign debt crisis in 2010. The prevailing, traditional pedagogical approach assumes that schools lead students to academic and vocational further education on the grounds of their academic performance. Over time, the ESF NOPs have stimulated the regions to broaden the scope of career guidance and vocational education and training (Colombo & Santagati, 2016). However, the region of South Tyrol and Trentino-Alto Adige is the only one to have institutionalized an encompassing system of VET pathways that resembles the large apprenticeship system that prevails in neighboring Austria. Between 2014 and 2018, the approach was rather sketchy in the other regions [46].

Over time, the focus of the official policies has evolved from instruction to competences and learning environments (e.g., "School for Development", 2000–2006; "Competences for Development", 2007–2013; "For the School. Competences and Learning Environment", 2014–2020). A common theme of these policy cycles has been that teachers were committed professionals that carried out their mission even in adverse circumstances. The logic of this policy strongly relies on the capacity of individual schools to improve the competences of at-risk youngsters by attuning evaluation systems to learning environments and fostering work-based learning [47].

In Liguria, "Competences for Development" became the basis of innovative but vague multilevel arrangements in the areas of vocational education and career guidance:

The Youth Plan left academic high schools [*licei*] in their ancestral situation, made up the mission of technical high schools [*istituti tecnici*], and eventually curtailed on-the-job training in professional schools [*istituti professionali*]. It remains unclear whether education and training is targeted to either further education or employment and whether either the central state or the regions must be accountable (Interview with a policymaker, in [42]).

Our interviewees were teachers and students from two professional high schools who had taken advantage of this ambiguity to pilot an alternative approach. Some teachers showcased their experiments with student-centered pedagogies and outdoors education in deprived neighborhoods, where they expected cooperative learning to make a difference:

Weak motivation is not only the consequence of the school curriculum but of the teaching method. A cooperative method transforms the more gifted students into the mentors of their classmates (Interview with a teacher, [42]).

Although the principal did not allow to do it during school time, I started to meet a mixed ability class outside the school every Wednesday. A school counsellor helped us to find a place in a local hostel. There we ate pizza, played music and the students talked to other local people. At the end, we produced a film plot including the pictures of everybody, from the twelve- year- old student to the older people around (Interview with a teacher, in [42]).

The local Music House Cooperative also facilitated an arena for alternative learning based on the arts:

It's not only instruction but also other types of suffering that provoke early leaving. In our project, interest on music is the response to the risk of dropping out today but also to other forms of deviation tomorrow (Interview with a leisure educator, in [42]).

Many participants in the focus groups blamed teacher-centered pedagogies for their previous disengagement from school life. In their view, this method was inefficient and outdated, not least because it overlooked the real sequence of their individual learning and their ulterior improvement. They thought that teacher-centered pedagogy is unhelpful for the current challenges of work and jobs. On the contrary, they praised the potential of new instruments that their teachers were developing, such as narrative and creative laboratories in the school, much needed career guidance and flexibility to move from one specialty to another one. They stated that they felt increasing confidence in their own abilities.

Frontal lessons don't work, the relationship between teachers and students must be circular and more engaging (Interview with a student, in [42]).

In lower secondary schools there is no guidance... if you are not good at school, they send you to vocational schools (Interview with a student, in [42]).

It should take more hours of manual practice to be an electrician (Interview with a student, in [42]).

It was useful when we did the group work, the afternoon study support and the sport activities... but that project lasted too short (Interview with a student, in [42]).

Local policymakers challenged rigid classifications of school outcomes according to students' age:

All of us, State [i.e., central government], regions and schools have to acknowledge that between the ages of 14 and 17 the youths live in a frontier where they must find the pathway. We must remember that these students do not only need vocational training but also citizenship skills that will help them to be professionals in some years' time (Interview with a policymaker, in [42]).

Afterwards, the regional Three-Year Pathways Program broadened the time frame that teenagers have to achieve middle qualifications in water resources management, mechanical and electrical engineering and the hospitality sector. By collaborating with the employers in key industries of the regional economy, these pathways are expected to reduce pressure for academic credentials in the short term and provide an opportunity to complete compulsory education and simultaneously deliver on-the-job training [48].

Teacher interviewees blamed most youth programs for focusing on short-term events like completing a course or finding a job, thus overlooking the wider meanings of the students' biographical experiences.

> Our school works in a frontier for multiple reasons. This neighborhood was built as an industrial town in the sixties, but afterwards suffered from severe economic decline. So, we must be aware of personal aspects, but we cannot reduce the problem to these aspects. It is important to recognize the effects of territorial segregation, of economic crisis and of unemployment (Interview with a teacher, in [42]).

> Since these children cannot envision a future, they do not see the point of doing what the school requires. We have a narrow margin of choice. Most problems erupt all of a sudden (Interview with a teacher, in [42]).

Teachers also noticed that informal interactions between teachers, students and other staff spaces during the school day had become a valuable pedagogic resource:

> To the extent that they talk to the janitor ten minutes each day, they feel they get in touch with real people. That is not possible in too rigid schools. However, when they are free to talk to who they want in the very school, they find the socialization, the love, the calm they need in order to prepare for the next class (Interview with a teacher, in [42]).

Student interviewees wished to transform the school into a space of dialogue where teachers, current and former students could build a common vision. Less formal relationships, interactive methodology and a flexible use of local facilities during the day were also important for the youth. They emphasized that many lectures were not spaces of communication:

> During the lectures, teachers must follow the syllabus and are not interested in us and do not listen to us and there is no interaction (Interview with a student, in [42]).

They praised innovative attempts to introduce mentoring and outdoors education in their school:

> The peer education and the tutoring projects allowed the enhancement of our skills through shared study (Interview with a student, in [42]).

> The open school in the afternoon is the real school because we know each other and we talk (Interview with a student, in [42]).

> The school does not have to be a place, but it is everywhere: in museums, in the streets, in bars, in the meadows. . .. It's a way of thinking (Interview with a student, in [42]).

In sum, two vocational high schools attempted to respond to the needs of students who were at risk of leaving without any academic credential. For such purpose, these schools crafted their own multilevel arrangement, wanted to widen the reach of student-centered pedagogies and experimented with more open and permeable concepts of school time. By doing so in accordance with a few community associations, these schools had helped students to be more assertive despite living in conditions of social vulnerability. Thus, these schools increasingly acted as collective agents even if, given the temporality of the projects, their impact remained uncertain in the middle term. The students were satisfied with the

projects that enhanced their soft skills but complained that this type of program was too short [42,49].

## 6. Governing Educational Pathways in Catalonia (Spain)

Despite ideological cleavages, in Spain, most central and regional educational authorities have converged in tackling disadvantages in the last years of compulsory lower-secondary education. From when a comprehensive school act was launched in the nineteen-nineties until a further reform openly challenged the hierarchy between academic and vocational tracks in 2020, policymakers and teachers assumed that single-ability grouping of (at least) 14- and 15-year-olds would eventually help low-performing students to fit into school life [50].

In Catalonia, the municipality of Barcelona shares the governance of the school system with the regional government in the city. The municipal Educational Services have invited high schools to change their own institutional culture to deploy holistic strategies that mitigate the risks of early school leaving [51]. The Barcelona Provincial Authority [52] supports municipalities to carry out their responsibilities in the domain of education and training. This authority has funded a few non-profits to pilot the concept of "second-chance education" that outlines an alternative type of school. Significantly, the officers and the participants in these two initiatives often reiterate their interest in adopting ideas espoused by the European Union [51].

The Catalan Employment Service has designed a few programs that cater to the unemployed youths, who are often not in education and training. The Employment Service funds municipalities and (often local) non-profits to deliver guidance and training for one year at most, but a few initiatives have attempted to broaden the scope of action. Since 2015, the Employment Service has funded partnerships of non-profits that run two-year initiatives in several localities (Government of Catalonia, 2018). In 2017, in Girona and other local areas, the county (*comarca*) piloted a common information service that attempted to coordinate training programs targeted to the youth in several neighboring towns.

> It was the first time we coordinated our work across the county. Although the whole county at most is like a district in Barcelona, previously each municipality exclusively focused on the youth that were registered there. Even worse, for some professionals it is necessary to compete with any other initiative that may be appealing for 'their' youth, thus reducing their budget. Working together has really been a nice opportunity (Interview with a policymaker, in [45]).

These programs normally take the interests of the student into account. Albeit blurred terms, the institutional vision and the opinion of the educators clearly contrast with the more academic pedagogies that most secondary schools use to teach high- and average-ability students. Often, these professional interviewees adhered to holistic, encompassing approaches that deployed career guidance to foster students' self-esteem and encourage them to elaborate more ambitious prospects [43,50].

Like similar students in Liguria, students of second-chance schools participating in some focus groups in the Barcelona metropolitan region openly reflected on how they had felt uncomfortable with teacher-centered pedagogy. Many attributed their previous disruptive behavior to derogatory comments of their teachers in compulsory education. In contrast, they praised personalized teaching and formative evaluation that takes their progress into account [43]. In Girona, the youth who participated in two-year programs sponsored by the Catalan Employment Service also remembered similar school experiences and recognized the importance of good relationships with educators. These slightly older interviewees also referred to the value of long-term approaches that linked education, guidance and leisure activities [45].

As a rule, students were particularly aware of pedagogical disparities between formal schooling and non-formal special programs. Many were very satisfied to know about the multiple dimensions of learning:

Maybe a person is much better at communicating with her hands that with words. Do you understand me? I am like that. I do practical tasks better than sitting to read a book and behave (Interview with a student in a second- chance school, in [43]).

The differences between the [official] secondary school and the [special, non-formal] second- chance school are how they treat you, their understanding and patience. It is about equality (Focus group with students in a second- chance school, in [43]).

We chose the picture where all students get to the finish line. Each one reaches his own, because you decide which is your finish line (Focus group with students in a second- chance school, in [43]).

They claimed they had become much more assertive after this alternative experience. Then, they felt like making more assertive decisions thanks to this pedagogic approach.

I will undertake vocational training in mechanics in the second- chance school. My current [special] program illuminated my way, or maybe opened doors (Interview with a student in a second- chance school, in [43]).

Finally, I chose to feel well at school and go there to work. If you do not feel well, you do not feel like studying. You don't do anything. You will do the same every day until they get rid of you (Interview with a student in a second- chance school, in [43]).

Here they know how to deal with me. I have been well treated. The led me to the right way. I learnt. It is a good experience (Interview with a student in a vocational training program, in [45]).

These interviewees challenged the standard construal of school space and time in several ways. Their student-centered pedagogies broadened the array of the future opportunities that students perceive. In Barcelona, a second-chance school had inspired the following reaction.

I would choose the picture of the teacher who opens the window. She opens it so that we can leave and follow our way. Sometimes you have the window, but they slam it on your face. Teachers say that you are worthless. They open another way that may be good for them, but it is f… difficult for you (Interview with a student in a second- chance school, in [43]).

In Girona, a young beneficiary of the two-year program accounted for his experience in these terms:

If I find my real vocation, it is much better. But I want to try before I make any decision. Things are either nice or ugly from outside, but everything changes when you are inside. When you do something every day, you really learn if it is interesting for you. Of course, you may also realize it is boring. The point is you must find the middle ground (Interview with a student in a vocational training program, in [45]).

The vision of the Barcelona Education Transitions Program endorses a transformation of school times quite explicitly:

In order to cope with such emotional distress, we thought of offering a key leverage of our methodology to our students: a different look, a different space and a different time [52].

The following excerpts from interviews with a student and an educator in Girona illustrate how time makes a big difference.

I always go to the Youth Point to ask for suggestions. I wanted to have some further education, but I was sent to take it part-time in the adult school, because

they did not recognize my qualifications from Senegal. However, now I have up-to-date information on both courses and jobs if I go to the Youth Point regularly, because they are aware (Interview with a student in a vocational training program, in [45]).

When we started a new (shorter than one year) program, the Catalan Employment Service required us to recruit the students among the list of unemployed youth that they provided. However, those lists were often incomplete, and many times duplicated the references. Since the same professional team has run the same program for five years, now we have our own pool of local youth. We are continuously in touch with them through WhatsApp. So, we can easily find out who is interested and contact them in time (Interview with an educator in a vocational training program, in [45]).

In Catalonia, some spurs of multilevel coordination between regional, provincial and local authorities have induced a few educators to experiment with innovative policies that use student-centered pedagogies and elaborate diversity-friendly images of students' futures and rhythms of education. Small networks of local services, non-profits and schools have become collective agents that vindicate this approach. But, it is hard to predict their impact in the middle term.

## 7. Comparative Analysis

According to our evidence, initial VET schools have adopted a broad concept of the right to education in some localities in Liguria (Italy) and Catalonia (Spain). Instead of a fixed-term commitment with the education of minors, these schools have bridged school and further education in innovative ways. The teachers have become policy actors who learn to deal with the consequences of early leaving from a fresh perspective by navigating the guidelines, the contradictions and the changes in European and national policies.

Two features of these schools stand out. On the one hand, they have drawn on local coalitions to deal with early school leaving regardless of the mainstream approach in the country. On the other hand, these schools have institutionalized a mode of temporality that articulates academic years and employment prospects with the multi-sided interactions of everyday life inside and outside the school.

In the mid-2010s, some Italian vocational schools changed their pedagogical approach and their everyday routines to cater to the needs of at-risk students. The ambiguity of official policies on the connections between education and employment eventually led the teachers to search for alternative pedagogies by themselves. To do so, they realized that they had to teach students out of the physical place of the school in such community venues as the youth hostel, the opera house and a museum. Students reacted very positively to these practices that broke the school routines that they had previously known. A few years later, the regional government also decided to approach schools and local communities by means of longer-than-usual initial VET pathways that responded to previous analysis of the needs of local employers.

In Catalonia, since 2010, the Municipality of Barcelona has gradually induced secondary schools to use student-centered, comprehensive pedagogies. The Barcelona Provincial Authority has funded non-profits to run second-chance schools. The regional government collaborates with local authorities in deploying monthslong vocational training; but since 2015, they have experimented with longer initiatives that either deliver career guidance or contact the youth through local informal networks. Therefore, the educators involved in these initiatives have learnt to use student-centered pedagogies for young students, to build networks of stakeholders and to welcome feedback given by their students and beneficiaries on the grounds of their own experience.

After years of practice, these schools and programs have institutionalized some types of multidimensional temporality that frame the sequence of courses and assessment within a broader context of everyday experience in which teachers and students easily engage in

discussions about the future. Certainly, these experiments will provide quite telling lessons for ulterior lifelong learning policies.

In our two Ligurian schools, some years ago, the teachers already realized that the mainstream linear temporality alienated many students. They noticed that these youth were much more motivated if they participated in very open discussions of future prospects. The possibility of having informal chats with the janitors also proved to be instrumental for their engagement. In our interviews, the students welcomed opportunities to interact with their peers in the class and indicated that the school was open in the afternoon. Interestingly, a student defined education as a "way of thinking" that students can apply to their activities in schools and other places in the community.

In Catalonia, not only student and teacher interviewees but also some official policy designs explored concepts of time that embed academic activity in everyday life and recognize the prospects of students. Therefore, these schools and training programs have learnt to link teaching with discussions of the future, to facilitate activities other than formal lessons in the very space of teaching and learning, and generally, to think of education beyond the walls and timetables of a school.

This analysis contributes to previous theoretical reflections on education, space and temporality. We use multilevel research to spell out very relevant clues of geographical scaling, local coalition building and the institutionalization of school temporality. In Italy and Spain, our analysis suggests that the incipient (albeit weak) influence of lifelong learning approaches to early school leaving depends on configurations of social relations that are not fixed but change depending on how policy actors structure opportunities in particular settings. Young people take advantage of the opportunities that they find near their dwelling. Their agency is indispensable for them to make sense of education and training in the frame of their life plans. But, they do not simply choose opportunities from a standard catalogue. On the contrary, they feel that some education and training programs are very helpful if they can frame this guidance and training in a multidimensional everyday life that is open to the experiences of diverse people.

## 8. Conclusions

Sociology has contributed to understanding youth transitions by looking at both the institutional arrangements that define certain pathways and the subjective construal of young people who either undertake or drop out from these pathways. This strand of research has found crucial aspects of space and temporality. To be precise, it has documented how a variety of policy actors have built modes of support and social representations of the right to education beyond the school leaving age. So far, the literature distinguishes groupings of countries that organize such support differently.

However, the literature also suggests that local policymaking is at stake. Some pieces of research show that the Scandinavian enabling model of youth citizenship and the Central European employment-centered mode ultimately rely on local arrangements between several public services and providers of education and training. Previous studies have also shown that narrow official understandings do not preclude vocational schools to bridge compulsory and further education in ways that empower disadvantaged young people.

In this article, we have explored examples of such local exceptions in two countries, Italy and Spain, in which mainstream policies exclusively deliver remedial support to the most disadvantaged and downplay the agency of young people. By reviewing some pieces of previous research, we have argued that a few schools can become more effective through alliances with some local stakeholders. At the same time, we have pointed out that these schools are quite skillful at embedding multi-sided, multidimensional routines into the frame of school-like slots of teaching and free time. Teachers and students themselves are capable of institutionalizing pedagogical and institutional approaches that depart from the official perspective.

Although we expect that this contribution will be fruitful for further research, certainly we are aware of both the scientific and the practical limitations of our study. On the

one hand, the evidence from Liguria and Catalonia has found counterintuitive exceptions to the prevailing pedagogical and policy approaches to further education and youth work in Italy and Spain. These exceptions strongly suggest that teachers and students can build trust even amid extremely narrow official policy designs. However, this point is only a small step forward. We need further research to understand the complex interaction between the European, the national and the local layers of policymaking around education and training.

On the other hand, the insistence of the official discourse of the European Union on multilevel governance and partnerships has some value, mostly due to the variety of local, informal and creative initiatives. Teachers and officers have widely proved their capacity to innovate by collaborating in networks. Notably, the young beneficiaries of many programs addressed to reduce early school leaving and open new opportunities for the disadvantaged youth normally send a clear message by reflecting on their own learning as a growth process. Good intentions notwithstanding, these policies are unlikely to help these disadvantaged youth if decisionmakers adopt a top-down approach that privileges benchmarks and accountability at the expense of participation. Our point is that, in order to be successful, this policy must become permanent and ongoing by supporting local coalitions and instill multidimensional temporality into school routines.

**Author Contributions:** M.B. conducted research in Liguria, while X.R. participated in several research projects on the topic in Catalonia. Both of them collaborated in the re-analysis of that material and wrote this manuscript. All authors have read and agreed to the published version of the manuscript.

**Funding:** Bartolini was funded by Università degli Studi di Genova Decreto 1111 (22 March 2016). Rambla et al. (2018) received funding from the European Union's Horizon 2020 research and innovation pro-gramme under grant agreement No 693167.

**Institutional Review Board Statement:** Bartolini was not obliged to apply for ethical clearance by her institution. Rambla et al. (2018) followed the instructions established by the UAB Research Ethics Committee Resolution num 3589 (30 March 2017).

**Informed Consent Statement:** Informed consent was obtained from all subjects involved in the study.

**Data Availability Statement:** Data are not available due to privacy reasons.

**Conflicts of Interest:** The authors declare no conflict of interest.

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
