# Peer review of "An Institutional Analysis of Local Lifelong Learning Approaches to Early School Leaving in Italy and Spain"

_societies, doi:10.3390/soc13110230_

Round 1

Reviewer 1 Report

Comments and Suggestions for Authors

Lack of clearly defined methodological assumptions.

The text needs improvement!!!

English is incomprehensible!!

Comments on the Quality of English Language

Lack of clearly defined methodological assumptions.

The text needs improvement!!!

English is incomprehensible!!

Author Response

Dear reviewer

Thank you for your comments.

We reviewed the wording to make a more comprehensible version. 

We also clarified some points in the methodological section (Taking geographical levels into account). Basically, we explore similar 'exceptions' to national narrow understandings of the right to education in Italy and Spain. We focus on two regions that reproduce the general institutional arrangements and statistical features of early leaving from education and training in these countries. To do so, we re-analysed previous studies that capture the views of policymakers, teachers and students. 

Regards

The authors

Reviewer 2 Report

Comments and Suggestions for Authors

The article presents significant findings that illuminate the efforts and constrained agency of local actors within educational structures. I think the following points would allow the paper to convey these points more clearly and appeal to a broader audience. The discussion of temporality is particularly compelling and the results convincing but the link to “denied citizenship” was not as useful nor enriching given how little was devoted to explaining the concept.

  • The article should more clearly explain the link between modes of youth citizenship and educational regimes. For instance because we know that the dual system in Germany/Austria etc compensates (to some degree) the inequalities entrenched by early tracking, while the FE and VET system in the UK seemingly entrenches disparities. Comments on 307-310 Would have made more sense if these regimes and their effects set out before. In general, more space should be devoted to explaining the broader policy context. For instance, what characterizes employment-driven regimes? Any details of the policy context for an audience that does not know? The article glosses over these explanations and that meant that at times during the article I found myself wondering, what exactly is “denied citizenship”? What are enabling and the monitored modes of youth citizenship? The paper risks being too specialist. Another example of this is the use of references that remain somewhat unexplained if somebody is not already well-acquainted with the literature (e.g. Walther’s (2017) and Chevalier’s (2016) typologies). 
  • Policy context only given 268-273 and very minimal. Could authors better set the scene  with some facts concerning school leaving to contextualise Italy and Spain in comparison to other countries? A discussion should also be possibly included in terms of returns for education as these are, for instance, lower in Italy than in, say, the UK and perception of these lower returns can shape engagement in FE and HE. I would also have expected more in terms of discussing the relationship with NEET policy (first mentioned at 441). Finally, the relationship between FE and HE that is briefly mentioned late in the paper should be discussed in conjunction with the policy context.
  • The emphasis on the role of local actors is illuminating but it would be more effective against a backdrop of the policy context and institutions, which is lacking as per comments above. In this sense, there is also little mention of accountability mechanisms characterising the system, bureaucracy constraints etc and these matter to understand the spaces of freedom afforded to schools and teachers. There should be a point in reflecting how school autonomy promoted in Italy gives freedoms but also shifts responsibility to schools that are remedying, in the limited local, way possible for them, to disadvantages that are entrenched in the system. If these were better fleshed out we could further appreciate not just the existence of the cleavages that the article helpfully highlights, but also the limitations in the specific systems.
  • The conclusion discussing limitations and their assessment is fair, and useful to point towards future paths for research at EU level beyond the 2 countries 

Author Response

Dear reviewer 

Thank you for your comments.

We realised that the first version did not clarify the relevance of the key institutional arrangements because we had relied too closely on Chevalier's (2016) modes of youth citizenship. For this reason, we attempted to clarify the main institutional features of the youth transition regime in Italy and Spain instead of using that typology so directly. 

The idea is exploring local exceptions in which VET schools draw on lifelong learning approaches although the mainstream national approach disregards this concept. Chevalier (2016) distinguishes modes of youth citizenship that articulate economic and social support differently. The two countries are 'extreme' cases in which these types of support are very weak. Then, we review previous research in a 'standard' region that reproduces the main institutional arrangements and statistical trends of the whole country.

We find out that some VET schools have been capable of catering to disadvantaged students by bridging the rationale of compulsory education with the premises of lifelong learning. In essence, the teachers of these schools have built local alliances and organized school time taking both academic knowledge and students' experience into account.

These local 'exceptions' complicate the concepts of agency and structure that the literature on youth transition regimes has adopted. At the same time, they notice some potential for change and indicate strategies to tackle early leaving.

Regards

The authors

Reviewer 3 Report

Comments and Suggestions for Authors

First of all, congratulations to the authors of the article, since they address a key issue within the challenging processes in education worldwide. Searching for solutions to resolve the issue that is of great concern, such as the trajectories of secondary education students, as well as the analysis of public policies, is essential. Describing how schools respond to this situation of early leaving, in countries like Spain and Italy, is particularly significant. The research questions are coherent, first: To what extent can schools create political coalitions at the local level? The review of other studies indicates that negotiation and learning cause changes. The next question: Are VET schools and programs capable of adapting to other times when young people can be more reflective, know the opportunities that await them, and become protagonists of their decisions? To do this, a local network space is also configured, experiencing a multidimensional temporality. The inquiry procedure is governed by the general guidelines of the comparative method, a multidimensional and multilevel design. Interviews were designed and focus groups were used. The analyses carried out contribute to theoretical reflections on education, educational space, and time. Now, the evidence that they have been able to reflect in this research contributes to the incorporation of pedagogical and political approaches in higher education and youth work in Italy and Spain.

Author Response

Dear reviewer

Thank you for your comments.

In order to tackle the issues raised by the whole set of reviewers, we submit a second version that has changed three aspects of the initial one:

  1. We have edited an English draft thoroughly.
  2. We have clarified some issues concerning the comparative method and the relevance of a re-analysis of previous research.
  3. We have described the key institutional features that underpin the significance of Italy and Spain for analysing policies against early leaving from education and training.

Regards

The authors

Round 2

Reviewer 1 Report

Comments and Suggestions for Authors

I recommend the text for publication!